Regulation of lymphoma in vitro by CLP36 through the PI3K/AKT/CREB signaling pathway

Lv Chao 1
Chen Guannan 2
Lv Shuang 1 lvshuang@163vip.com
1 Medical Oncology, Inner Mongolia People’s Hospital , Hohhot , China
2 Hepatological Surgery Department, Tianjin First Central Hospital , Tianjin , China
Guan Fanglin
Electronic publication date: 2024 Dec 24
Publication date: 2024
Volume: 12
Electronic Location ID: e18693
Received 2024 Sep 10; Accepted 2024 Nov 20
Copyright: © 2024 Lv et al.
Copyright year: 2024
Copyright holder: Lv et al.
License: This is an open access article distributed under the terms of the Creative Commons Attribution License, which permits unrestricted use, distribution, reproduction and adaptation in any medium and for any purpose provided that it is properly attributed. For attribution, the original author(s), title, publication source (PeerJ) and either DOI or URL of the article must be cited.
License URL: https://creativecommons.org/licenses/by/4.0/

Keywords: CLP36, Lymphoma, Proliferation, Survival, PI3K/AKT/CREB

Funding: The authors received no funding for this work.

==============================
Background

CLP36 is also known as PDZ and LIM Domain 1 (PDLIM1) that is a ubiquitously-expressed α-actinin-binding cytoskeletal protein involved in carcinogenesis, and our current study aims to explore its involvement in lymphoma.

Methods

Accordingly, the CLP36 expression pattern in lymphoma and its association with the overall survival was predicted. Then, qPCR was applied to gauge CLP36 expression in lymphoma cells and determine the knockdown efficiency. The survival, proliferation and apoptosis of CLP36-silencing lymphoma cells were tested. Cell viability, proliferation and apoptosis were assessed based on cell counting kit-8 (CCK-8) assay, colony formation assay, EdU staining, and flow cytometry, respectively. Additionally, qPCR was used to calculate the expressions of proteins associated with metastasis and apoptosis, while immunoblotting was employed to determine the phosphorylation status of phosphoinositide 3-kinase (PI3K)/protein kinase B (AKT)/cAMP-response element binding protein (CREB).

Results

CLP36 expression was relatively higher in lymphoma, which was associated with a poor prognosis. Also, CLP36 was highly-expressed in lymphoma cells and the silencing of CLP36 contributed to the suppressed survival and proliferation as well as the enhanced apoptosis of lymphoma cells. Further, CLP36 silencing repressed the expressions of Cadherin 2 (CDH2) and Vimentin (VIM) yet promoted those of Bax and Caspase 3 in lymphoma cells, concurrent with the reduction on the phosphorylation of PI3K, AKT and CREB, all of which were confirmed to be positively correlated with CLP36.

Conclusion

This study, so far as we are concerned, provided evidence on the involvement of CLP36/PI3K/AKT/CREB axis in lymphoma, which may be contributive for the identification on the relevant molecular targets of lymphoma.

Introduction

The recent World Health Organization (WHO) classification, which was revised in 2017, has comprised over 80 types of mature lymphoid neoplasms (Hodgkin lymphomas, B-cell, and T-cell) (de Leval & Jaffe, 2020). Lymphoma is one of the most common forms of hematologic malignancies originating from the lymphopoietic system and represent 4% of novel malignancies in Western countries, ranking the fifth in the most prevalent cancer and the major cause leading to cancer mortality (Asgaritarghi et al., 2023; Zhuang et al., 2024; Elenitoba-Johnson & Lim, 2018). Nonetheless, patients with lymphoma often fail to respond to treatment or relapse early even under the current treatment standards (Yu et al., 2023). Genetic mutations have been shown to partly activate inflammatory pathways and modulate the upstream or downstream targets of pathways in lymphoma (Shi et al., 2024). Our current study, accordingly, focuses on the molecular mechanisms of the relevant molecular biomarkers and corresponding signaling pathway in lymphoma to offer a reference for the development of effective therapeutic regimens.

Increasing evidences have recognized the relevance of epigenetic alternations in the pathogenesis of lymphoid malignancies, with some agents evaluated in clinical studies (Booth & Collins, 2021). Cytogenetics can be of help in simplifying the diagnostic complexities presented in transforming and progressive lymphoid malignancies (Dave, Nelson & Sanger, 2011). Further, gene expression profiling has provided a quantitative molecular framework in the study of human lymphomas (Shao et al., 2024; Staudt & Dave, 2005). For instance, the deregulation of MYC, one of the most deregulated oncogenes in human malignancies, has been already characterized (Sewastianik et al., 2014). In the meantime, a transcriptional repressor, BCL6, has been also recognized as a therapeutic target for lymphoma (Leeman-Neill & Bhagat, 2018). Further, the adaptor protein human germinal center-associated lymphoma (HGAL), which is specifically expressed in germinal center lymphocytes, can repress lymphoma dissemination through interaction with multiple cytoskeletal proteins (Jiang et al., 2021).

CLP36 is a member of the ALP/Enigma protein family showing widespread distribution in diverse non-muscle tissues, which contains 1 PDZ domain at the N-terminus, 1 LIM domain at the C-terminus and a ZASP-like motif between the PDZ and LIM domains (Miyazaki et al., 2012; Gupta et al., 2012; Lu et al., 2022). As a crucial player in cytoskeletal organization and organ development, CLP36/PDLIM1 has been recently demonstrated to be dysregulated in a variety of cancers with important roles in the proliferation and metastasis during tumor initiation and progression (Zhou et al., 2021). While trying to link CLP36/PDLIM1 with malignancies, it should be noticed that CLP36 is involved in the apoptosis and proliferation of chronic myelogenous leukemia cells (Li, Luo & Song, 2020). In the meantime, CLP36 was shown to interact with p75 neurotrophin receptor (NTR) in highly invasive patients-derived glioma cells and short hairpin RNA-mediated CLP36 knockdown led to the complete ablation of p75(NTR)-mediated invasion (Ahn et al., 2016). Besides, CLP36 enhanced the migration, invasion and metastasis of breast cancer cells via interacting with α-actinin (Liu et al., 2015). Additionally, a preliminary investigation has suggested the involvement of PDLIM family in acute myeloid leukemia, another aggressive hematological malignancy (Cui et al., 2019). These evidences have made us curious whether CLP36 was involved in the malignant behaviors of lymphoma cells, and this study hence commenced for the validation.

Therefore, our study aimed to investigate the role of CLP36 in lymphoma by regulating the PI3K/AKT/CREB signaling pathway. We used bioinformatics analysis, cell culture and molecular biology experiments in order to confirm that CLP36 expression in lymphoma cells correlates with disease prognosis. In addition, lymphoma cell proliferation, apoptosis, and effects on PI3K/AKT/CREB pathway activity were observed by knocking down CLP36. In conclusion, our findings may provide new targets and perspectives for the treatment of lymphoma.

Materials and Methods

Bioinformatics

The CLP36 expression pattern in lymphoma and the association between CLP36 expression and the survival of lymphoma patients were predicted based on relevant data from the cancer genome atlas program (https://www.cancer.gov/ccg/research/genome-sequencing/tcga) (Shahrajabian & Sun, 2023). The correlation between CLP36 and PI3K/AKT/CREB was also determined.

Cell culture and intervention

Human lymphoma cell lines, including B-cell lymphoma cell lines WSU-DLCL2 (ACC 575), SU-DHL-8 (ACC 573), OCI-LY19 (ACC 528), SU-DHL-2 (ACC 902), and Burkitt lymphoma cell lines BJAB (ACC 757) and RA 1 (ACC 603) were all ordered from Deutsche Sammlung von Mikroorganismen und Zellkulturen GmbH (Braunschweig, Germany) and cultured as follows. For B-cell lymphoma cell lines, 90% Roswell Park Memorial Institute-1640 medium (90023; Solarbio LifeSciences, Beijing, China) or α-minimal essential medium (M0644; Millipore Sigma, Darmstadt, Germany) with the supplementation of 10% fetal bovine serum (12103C; Millipore Sigma, Darmstadt, Germany) was applied for the culture. Meanwhile, 80% Roswell Park Memorial Institute-1640 medium and 20% heat-inactivated fetal bovine serum (12106C; Millipore Sigma, Darmstadt, Germany) were used for the culture of Burkitt lymphoma cells. All cells were incubated at 37 °C with 5% CO2.

For the intervention assays, the small interfering RNAs (siRNAs) against CLP36 (si-CLP36#1, target sequence: 5′-TGGGGAAAATACTAGCAATATGA-3′; si-CLP36#2, target sequence: 5′-GTCATCCATACAAGATGAATTTA-3′) and the corresponding negative control with scramble target sequences (5′-ATTAGTGCAGATACATCTTACAA-3′) were all synthesized by GenePharma (Shanghai, China) and transfected into BJAB and RA 1 cells using Invitrogen™ Lipofectamine 2000 transfection reagent (11668027; ThermoFisher Scientific, Waltham, MA, USA) for 48 h, as recommended by the protocols of the producer.

Cell counting kit-8 assay

Cell viability of CLP36-silencing lymphoma cells was gauged with a commercial CCK-8 assay kit (CK04; Dojindo, Kumamoto, Japan). In detail, lymphoma cells BJAB and RA 1 were seeded in a 96-well plate at the density of 2 × 103 cells/well and cultured for 12, 24 and 36 h. Then 10 μL CCK-8 solution was added to the cells for another 4-h culture. IMark™ microplate absorbance reader (1681130; Bio-Rad Laboratories, Inc., Hercules, CA, USA) was used to measure the OD value at 450 nm (Ruan et al., 2024).

Cell proliferation assays

For colony formation assay, following the transfection, BJAB and RA 1 cells were grown in a 6-well plate at 1 × 103 cells/well and cultured at 37 °C for 14 days. Hereafter, these cells were fixed in methanol (M813896; Macklin, Shanghai, China) for 15 min and dyed using crystalline violet solution (C776042; Macklin, Shanghai, China) for 30 minu. The colonies formed were observed and photographed and the number was hereafter quantified.

For the second part, BJAB and RA 1 cells were seeded in 96-well plates at the density of 3 × 103 cells/well for overnight culture, followed by the incubation with 50 μM 5-ethynyl-2′-deoxyuridine (EdU, E796036; Macklin, Shanghai, China) reagent at 37 °C for 1 h. Further, DAPI solution (C0065; Solarbio LifeSciences, Beijing, China) at 0.1 μg/mL was adopted for nuclei dyeing for 20 min at room temperature and the results were finally observed in a fluorescence microscope (DM6B; Leica Microsystems, Wetzlar, Germany) (Wang et al., 2022).

Cell apoptotic assay

FITC Annexin V/PI Apoptosis Detection Kit (CA1020; Solarbio LifeSciences, Beijing, China) was applied in cell apoptotic assay. Following the transfection, the treated BJAB and RA 1 cells (1 × 106 cells in total) were suspended in the binding buffer and centrifuged to remove the supernatant. Then, these cells were sequentially added with the working solutions of both FITC Annexin V and PI (5 μL for each) and cultured for 10 min. Ultimately, the results were analyzed in a flow cytometer (FACSVERSE; BD Biosciences, Franklin Lakes, NJ, USA) and FlowJo 10 (FlowJo, LLC., Ashland, OR, USA) was utilized for data analysis.

Quantitative PCR

The total RNA from cells were extracted using Invitrogen™ TriZol assay kit (15596026; Thermo Fisher Scientific, Waltham, MA, USA) and then reverse-transcribed into cDNA using first-strand cDNA synthesis kit (11119ES60; Yeason, Shanghai, China). Hereafter, the PCR was initiated in a PCR thermocycler (CFX96; Bio-Rad Laboratories, Inc., Hercules, CA, USA) using the Hieff® qPCR Green Master Mix (11204ES03; Yeason, Shanghai, China) at 95 °C for 1 min and 40 cycles of 95 °C for 10 s, 60 °C for 20 s and 72 °C for 20 s. The relative mRNA levels of genes of interest were finally calculated with the method 2−ΔΔCt with GAPDH as the housekeeping control (Livak & Schmittgen, 2001; Sindhuja, Amuthalakshmi & Nalini, 2022). The primers used are shown in Table 1.

Table 1 Sequence of primers.

Gene	Primers (5′-3′)	
Forward (F)	Reverse (R)	
CLP36	GAGAAACAGGAGTTGAATGAGCC	GCAGCCACTTTAGTGACAGGAG	
CDH2	CCTCCAGAGTTTACTGCCATGAC	GTAGGATCTCCGCCACTGATTC	
VIM	AGGCAAAGCAGGAGTCCACTGA	ATCTGGCGTTCCAGGGACTCAT	
BAX	TCAGGATGCGTCCACCAAGAAG	TGTGTCCACGGCGGCAATCATC	
CASP3	GGAAGCGAATCAATGGACTCTGG	GCATCGACATCTGTACCAGACC	
GAPDH	GTCTCCTCTGACTTCAACAGCG	ACCACCCTGTTGCTGTAGCCAA	

Immunoblotting

In the present study, in order to further determine the expression of CLP36, as well as to assess the activation status of the signaling pathway, we performed assays using the western blotting assay. Cell lysates were prepared with the RIPA lysis buffer (20115ES60; Yeason, Shanghai, China) and the protein concentration was tested via BCA method. Following the separation in SDS-PAGE separation gel, the protein samples (30 μg) were transferred to the PVDF membrane which was hereafter blocked with 5% defat milk for 1 h at room temperature. PI3 Kinase p85 Antibody (#4292, 1:1000; Cell Signaling Technology, Danvers, MA, USA), Phospho-PI3 Kinase p85 (Tyr458)/p55 (Tyr199) Antibody (#4228, 1:1000; Cell Signaling Technology, Danvers, MA, USA), Akt Antibody (#9272, 1:1000; Cell Signaling Technology, Danvers, MA, USA), Phospho-Akt (Ser473) Antibody (#9271, 1:1000; Cell Signaling Technology, Danvers, MA, USA), CREB (48H2) Rabbit mAb (#9197, 1:1000; Cell Signaling Technology, Danvers, MA, USA), Phospho-CREB (Ser133) (87G3) Rabbit mAb (#9198, 1:1000; Cell Signaling Technology, Danvers, MA, USA) and GAPDH (14C10) Rabbit mAb (#2118, 1:1000; Cell Signaling Technology, Danvers, MA, USA) were applied for the overnight culture at 4 °C, followed by the further reaction with Anti-rabbit IgG, HRP-linked Antibody (#7074, 1:1000; Cell Signaling Technology, Danvers, MA, USA) for 1 h at room temperature. The visualization on the membrane was completed with the ECL visualization reagent (36208ES60; Yeason, Shanghai, China) and the densitometry on the protein band was determined with ImageJ 5.0 (Bio-Rad Laboratories, Hercules, CA, USA).

Statistical analysis

All statistical analysis of this study was performed using SPSS 17.0 (SPSS, Inc., Chicago, IL, USA) and all data of independent triplicate were expressed in the form of mean ± standard deviation. Differences among groups were compared by student’s t-test. Statistically significant was defined when a P-value was lower than the threshold of 0.05.

Results

CLP36 expression analysis in lymphoma and determination on the association with the prognosis

In the beginning, with the purpose of exploring the specific effects of CLP36 in lymphoma, we searched and downloaded the data of CLP36 expression pattern in lymphoma from The Cancer Genome Atlas program. Relevant data have suggested that CLP36 (shown as PDLIM1 in the figures) was highly expressed in lymphoma (Fig. 1A, P < 0.05), and such high CLP36 expression in lymphoma was related to an unfavorable overall survival of lymphoma patients (Fig. 1B, logrank P = 0.02).

Figure 1 Expression analysis of CLP36 in lymphoma and its association with the prognosis.

(A) Expression pattern of CLP36 (shown as PDLIM1 in the figures) in lymphoma based on the data from TCGA. (B) Association between CLP36 expression level and the survival of patients with lymphoma. *P < 0.05.

Effects of CLP36 silencing on the survival and proliferation of lymphoma cells

Then, the level of CLP36 in lymphoma cells was quantified accordingly, where a relatively higher expression was seen in BJAB and RA 1 cells as compared to that in WSU-DLCL2 cells (Figs. 2A–2C, P < 0.01). These two cells were hence applied in subsequent assays. Then, the level of CLP36 was forcibly knockdown in BJAB and RA 1 cells using the relevant siRNAs, and the successful transfection was witnessed, as indicated by the reduced level of CLP36 in these cells (Figs. 2D and 2E, P < 0.0001). The CCK-8 assay was applied to determine the viability of BJAB and RA 1 cells, as seen in the reduced OD value at 12, 24 and 36 h (Figs. 2F and 2G, P < 0.01).

Figure 2 Effect of CLP36 silencing on the survival of lymphoma cells.

(A) Relative CLP36 mRNA expression level in lymphoma cells quantified via qPCR. (B, C) Relative CLP36 protein expression level in lymphoma cells calculated based on immunoblotting assay. (D, E) Validation on the knockdown efficiency of CLP36-specific siRNAs (si-CLP36#1 and si-CLP36#2) on lymphoma cells BJAB (D) and RA 1 (E), as calculated by qPCR. (F and G) Effects of CLP36 silencing on the viability of lymphoma cells BJAB (F) and RA 1 (G) based on the results of CCK-8 assay. All experimental data of independent triplicates were expressed as mean ± standard deviation. **P < 0.01; ***P < 0.001; ****P < 0.0001.

The proliferation of CLP36-silenced lymphoma cells BJAB and RA 1 was evaluated based on colony formation assay (Figs. 3A and 3B) and EdU staining (Figs. 3C and 3D). Accordingly, it was visible that CLP36 silencing led to the reduced number of colonies formed (Figs. 3A and 3B, P < 0.01) and the number of EdU-positive cells (Figs. 3C and 3D, P < 0.05). Further, the levels of metastasis-related proteins cadherin 2 (CDH2) and vimentin (VIM) were calculated to evaluate the metastasis potential of lymphoma cells, and it was evident that the knockdown of CLP36 led to the diminished levels of these two proteins in lymphoma cells (Figs. 3E and 3F, P < 0.05).

Figure 3 Effect of CLP36 knockdown on the proliferation and metastasis potential of lymphoma cells.

(A, B) Effects of CLP36 silencing on the colony formation of lymphoma cells BJAB and RA 1. (C, D) Effects of CLP36 silencing on the number of EdU-positive lymphoma cells BJAB and RA 1. (E, F) Effects of CLP36 silencing on the proteins related to metastasis CDH2 (E) and VIM (F) in lymphoma cells BJAB and RA 1. All experimental data of independent triplicates were expressed as mean ± standard deviation. *P < 0.05; **P < 0.01; ***P < 0.001; ****P < 0.0001.

Effects of CLP36 knockdown on the apoptosis of lymphoma cells

The effects of CLP36 silencing on the apoptosis of lymphoma cells were further explored, revealing the elevated apoptosis rate in lymphoma cells BJAB and RA 1 following the knockdown of CLP36 (Figs. 4A–4D, P < 0.01). Such pro-apoptotic effects of CLP36 silencing on lymphoma cells BJAB and RA-1 were further confirmed by qPCR, which was applied to calculate the levels of relevant proteins Bax and caspase-3 (CASP3). Specifically, the silencing of CLP36 caused the elevation of Bax and CASP3 expressions in lymphoma cells BJAB and RA 1 (Figs. 4E and 4F, P < 0.05).

Figure 4 Effect of CLP36 silencing on the apoptosis of lymphoma cells.

(A–D) Apoptosis of CLP36-silencing lymphoma cells BJAB (A, B) and RA 1 (C, D), as quantified based on the flow cytometry assay. (E, F) Expression levels of BAX (E) and Caspase-3 (F) in CLP36-silencing lymphoma cells BJAB and RA 1, quantified via qPCR. All experimental data of independent triplicates were expressed as mean ± standard deviation. *P < 0.05; **P < 0.01; ***P < 0.001; ****P < 0.0001.

Effects of CLP36 silencing on PI3K/AKT/CREB pathway in lymphoma cells

Finally, we determined the downstream pathway underlying the effects of CLP36 silencing on lymphoma cells. The function of PI3K/AKT signaling pathway caught our attention, and the correlation between CLP36 and PI3K/AKT/CREB was determined firstly. Positive correlations were seen in CLP36 (shown as PDLIM1) and PI3K (Fig. 5A, P = 0.00066, R = 0.48), AKT (Fig. 5B, P = 0.04, R = 0.3) and CREB (Fig. 5C, P = 0.014, R = 0.36). Additionally, the phosphorylation level of PI3K, AKT and CREB was witnessed to be sharply diminished in lymphoma cells after the silencing of CLP36 (Figs. 5D and 5E, P < 0.05).

Figure 5 Effect of CLP36 silencing on PI3K/AKT/CREB pathway in lymphoma cells.

(A–C) Correlation between CLP36 and PI3K/AKT/CREB, as determined via Pearson’s correlation analysis. (D, E) Phosphorylation level of PI3K/AKT/CREB in lymphoma cells quantified via immunoblotting assay. All experimental data of independent triplicates were expressed as mean ± standard deviation. *P < 0.05; **P < 0.01.

Discussion

CLP36 (PDLIM1) is an actin-associated scaffold as a target hit and localizes to a variety of actin cytoskeletal elements, which is diffusely expressed in gastrointestinal tract, muscle, heart tissues and serves a cell-cell adhesion and focal adhesion modulator with functional importance in cancer (Dhanda et al., 2021; Chen et al., 2016). In our current study, so far as we are concerned, we firstly proved the oncogenic effects of CLP36 on lymphoma. In detail, following the confirmation that CLP36 was highly expressed in lymphoma cells, it was additionally observed that CLP36 silencing could lead to the suppression on the proliferation and metastasis potential of lymphoma cells and the enhancement on the apoptosis, which indirectly demonstrated that CLP36 may be associated with targeting the PI3K/AKT/CREB pathway (Fig. 6).

Figure 6 Action-of-mechanism of this study underlying the effects of CLP36 on lymphoma cells.

Cell proliferation has been demonstrated to be a crucial target in carcinogenesis, since cancer cells embody some characteristics that allow their survival beyond a normal life span and their abnormal proliferation (Lu et al., 2022; Feitelson et al., 2015). Tumor metastasis, in the meantime, has been defined as a multistage process where malignant cells spread from the original site and colonize distant organs or nodes, demonstrating heterogeneous biology (Nguyen, Bos & Massagué, 2009). Existing studies have implicated CLP36 as a pro-metastasis factor in breast cancer and glioma (Ahn et al., 2016; Liu et al., 2015). Considering the tissue specificity of CLP36, the specific role of CLP36 in tumor types may vary accordingly. Experimental and clinical studies have further demonstrated the gatekeeper role of epithelial-to-mesenchymal transition (EMT) in modulating tumor invasion and metastasis, where CDH2 and VIM have been underlined to be involved (Thiery et al., 2009; Zhang et al., 2023). CDH2 is a member of the cadherin family that regulates a variety of cellular processes like apoptosis, angiogenesis and chemoresistance and plays a profound role in the EMT (Gao et al., 2018). Meanwhile, VIM is an intermediate filament protein which plays crucial roles in EMT, as seen by the increased mesenchymal phenotypes (Ramos et al., 2020; Chen et al., 2024). While trying to link CLP36 and EMT, it has been demonstrated that CLP36 could prevent metastatic potential or EMT of colorectal cancer cells, as seen by the elevated CDH1 expression and the diminished VIM, Snail and ZEB levels (Chen et al., 2016; Kim et al., 2019). In the current study on the proliferation and metastasis potential of lymphoma, it was seen that CLP36 silencing could evidently suppress the proliferation and the expression levels of VIM and CDH2 in lymphoma cells. Further, some existing studies have further proven the involvement of Bcl-2 family proteins like Bax and caspase member CASP3 in apoptosis of lymphoma (Edlich, 2018; Eskandari & Eaves, 2022; Baas et al., 2022). Here, in addition to the confirmation that CLP36 silencing led to the aggravated apoptosis of lymphoma cells, it was seen that CLP36 silencing contributed to the elevation on the levels of Bax and CASP3.

Existing studies have demonstrated the potential of targeting some relevant signaling pathways in cancer therapy, like hypoxia-inducible factor-1 (Ma et al., 2021), Notch pathway (Hu et al., 2012), growth factor signaling (Lowery & Yu, 2012), PI3K/AKT pathway (Yu, Wei & Liu, 2022), MAPK pathway (Kciuk et al., 2022), and Wnt/β-catenin pathway (Wang, Li & Ji, 2021), to name a few. CLP36 has been demonstrated to mediate the Hippo-YAP signaling and Wnt/β-catenin pathways in gastric cancer (Tan et al., 2022; Lei et al., 2024). In our current study, we focused on the PI3K/AKT pathway, whose interaction with CLP36 has not been explored. The PI3K/AKT pathway is one of the most prevalent dysregulated signaling pathways in cancer patients, which exerts a critical effect on promoting tumorigenesis, progression and therapeutic response as well as the EMT process (Wang et al., 2023; Mo et al., 2023; Wei, Chen & Feng, 2023). PI3Ks constitute a lipid kinase family with the characterization of their capability to phosphorylate inositol 3′-OH group in inositol phospholipids, which can phosphorylate phosphatidylinositol-4,5-bisphosphate (PIP2) for the generation of PIP3 (Fresno Vara et al., 2004; He et al., 2021). PIP3 then recruits the oncogenic signaling proteins like the serine and threonine kinase AKT (Vanhaesebroeck et al., 2010). Hereafter, PI3K/AKT can trigger the phosphorylation of the downstream signaling regulator CREB (Yang et al., 2022). The efficacy of targeting PI3K/AKT in lymphoma and targeting CREB in hematopoiesis has been already addressed (Blachly & Baiocchi, 2014; Kinjo et al., 2005). In our current study, we confirmed that CLP36 silencing could diminish the phosphorylation of PI3K, AKT and CREB in lymphoma cells, thus indicating the targeting relationship between CLP36 and PI3K/AKT/CREB axis in lymphoma. Nonetheless, the involvement of PI3K/AKT/CREB axis in the proliferation and metastasis of CLP36-silenced lymphoma cells has not been expounded, which will become the goal of our future study for the validation. However, there are limitations to our study. Firstly, the number of patient samples used in this study was relatively small, and will be expanded in the future to include more lymphoma patients and a wider population, which will in turn strengthen the reliability of the findings. Second, although this study found that CLP36 was associated with survival in patients with lymphoma, its potential clinical applications were not fully explored. Finally, this study focused on the PI3K/AKT/CREB signaling pathway, and in the future, systems biology approaches, including proteomics and network analysis, will be used to comprehensively assess the signaling network involved in CLP36 and to identify other possible key regulatory pathways.

In short, we found that CLP36 expression correlates with the prognosis of lymphoma patients and that silencing CLP36 expression leads to reduced proliferative capacity and increased apoptosis of lymphoma cells, which correlates with inhibition of the CLP36/PI3K/AKT/CREB axis. These results implicate CLP36 and its associated pathways as potential therapeutic targets, which provide new perspectives and insights into the treatment of lymphoma.

Supplemental Information

Supplemental Information 1 All WB for original bands.

Supplemental Information 2 MIQE checklist.

Abbreviations

HGAL human germinal center-associated lymphoma

PDLIM1 PDZ and LIM Domain 1

NTR neurotrophin receptor

PI3K Phosphoinositide 3-kinase

AKT protein kinase B

CREB cAMP-response element binding protein

siRNAs small interfering RNAs

CCK-8 Cell counting kit-8

OD optical density

EdU 5-ethynyl-2′-deoxyuridine

q-PCR Quantitative PCR

CDH2 cadherin 2

VIM vimentin

CASP3 caspase-3

EMT epithelial-to-mesenchymal transition

PIP2 phosphatidylinositol-4,5-bisphosphate

PIP3 phosphatidylinositol-3,4,5-trisphosphate

Additional Information and Declarations

Competing Interests

Author Contributions

Data Availability

The authors declare that they have no competing interests.

Chao Lv conceived and designed the experiments, analyzed the data, prepared figures and/or tables, authored or reviewed drafts of the article, and approved the final draft.

Guannan Chen conceived and designed the experiments, performed the experiments, analyzed the data, authored or reviewed drafts of the article, and approved the final draft.

Shuang Lv performed the experiments, prepared figures and/or tables, and approved the final draft.

The following information was supplied regarding data availability:

The raw data (including flow cytometry data) is available at GitHub and Zenodo

- https://github.com/lvshuang176/Updated-all-data.git.

- lvshuang176. (2024). lvshuang176/Updated-all-data: Updated raw data1.1.2 (v.1.1.2). Zenodo. https://doi.org/10.5281/zenodo.14048696.

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
