# Peer review of "Regulation of lymphoma in vitro by CLP36 through the PI3K/AKT/CREB signaling pathway"

_PeerJ, doi:10.7717/peerj.18693_

## Round 0.1 · original submission · Major Revisions

While two reviewers suggested minor revisions, one reviewer recommended major revisions. After evaluating all the feedback, I concur that substantial changes are necessary to address the concerns raised and to strengthen your paper. Please carefully review all the reviewers' comments and address them comprehensively in your revision. Pay particular attention to the points raised by the reviewer recommending major revisions (R2), as these likely highlight significant points for improvement.

Reviewer 1 ·

Basic reporting

In this study, the author demonstrated that the CLP36 is a risk factor targeting the PI3K/AKT/CREB signaling pathway for lymphoma progression. The overexpression of CLP36 in lymphoma was associated with poor outcome, the silencing of CLP36 suppressed survival and proliferation of tumor cells, and affected the phosphorylation of PI3K, AKT and CREB protein in the PI3K/AKT/CREB signaling pathway. The experimental design of this study is reasonable, the argument is clear, the evidence is sufficient. In my view, this paper is meaningful for the study of the molecular mechanism of a and the development of potential therapeutic targets. However, the manuscript required further refinement before publication.
1. The risk factors (infection, immunity, physical and chemical factors) and clinical manifestation affecting the lymphoma, and the well-studied molecular mechanisms mediating the disease progression included what.
2. Line 51, the current treatment standards for lymphoma is what. How effective is the cure. The epstein-barr virus was reported to associate with lymphoma occurrence, what are the mechanisms by which these viruses induce the progression of lymphoma.
3. Line 51, Which genetic mutations are the key factors affecting the occurrence of lymphoma, including mutation type and mutation frequency. Which critical paths are included.
4. Line 58, Which biomarkers have been developed for the clinical diagnosis of lymphoma, and how effective is the diagnosis.
5. Line 81, Please briefly describe the research process and some important results in the last part of introduction.
6. Line 84, the CLP36 expression pattern affected the survival of lymphoma patients, Do the KM survival analysis were used. Has the relationship between CLP36 and PI3K/ AKT/ CREB been reported.
7. In the Figure 1. The CLP36 also termed as PDLIM1, but in others Figure, the name of CLP36 was is widely used. I think it would be more appropriate to replace PDLIM1 in Figure 1 with CLP36.
8. Figure 5, the figure legend showed the correlation between CLP36 and PI3K/AKT/CREB, please replace PDLIM1 with CLP36 in the Figure 5A, B, C. In addition, please check the significant marker of “*” in Figure 5E.
9. Line 198, the PI3K/AKT signaling pathway affected the tumor survival and metastasis. Please replace the sentence of “PI3K/AKT/CREB pathway caught our attention” with the function of PI3K/AKT signaling pathway reasonably.
10. What are the limitations of this study and implications for future research.

Experimental design

no comment

Validity of the findings

no comment

Reviewer 2 ·

Basic reporting

This study explored the role of CLP36 (PDLIM1) in lymphoma based on the cellular experiments. CLP36 expression was relatively higher in lymphoma, which was associated with a poor prognosis. Moreover, CLP36 was highly-expressed in lymphoma cells and the silencing of CLP36 contributed to the suppressed survival and proliferation as well as the enhanced apoptosis of lymphoma cells. Further, CLP36 silencing repressed the expressions of Cadherin 2 (CDH2) and Vimentin (VIM) yet promoted those of Bax and Caspase 3 in lymphoma cells, concurrent with the reduction on the phosphorylation of PI3K, AKT and CREB. As a whole, the experimental design of this study is rigorous, logical, and the writing is easy to understand. Whereas, there are still a few problems in the manuscript that requires to be resolved prior to publication.
1. In my opinion, the title of this article is suggested to be revised in a straightforward way that is better to express the crucial results or findings of the study.
2. In the Methods of Abstract section, the author needs to detailly describe each assay used in this study, such as cell counting kit-8 (CCK-8) assay for detecting the cell viability of CLP36-silencing lymphoma cells.
3. The examples "MYC", "BCL6", and "HGAL" listed in the second paragraph of the introduction seem to be not closely related to the main theme of this study, and it is recommended to delete them and integrate this paragraph with the third paragraph of "CLP36".
4. In consideration of the purpose of this study that reveals CLP36 targeting the PI3K/AKT/CREB signaling pathway in lymphoma, it is suggested to include more relevant literatures on the role of PI3K/AKT/CREB signaling pathway in the progression of cancer, especially Lymphoma, into the manuscript, so as to closely link the content of this study to provide a detailed background introduction.
5. At the end of the introduction, please add a paragraph to summarize the work to be carried out following, including the major methods and research significance.
6. Please supplement the specific details on which TCGA dataset is based on to perform the bioinformatic analysis, and the patient information involved (sample size and survival time of patients, etc).
7. What is the purpose of Immunoblotting experiments? Please laconically state it in the corresponding part of the manuscript.
8. Although this study conducted many experiments and found some important results, there are still several limitations. Thus, it is suggested to elaborate the prospects and shortcomings of this study in the discussion section, as well as its guiding significance for subsequent research work.
9. The caption of Figure 1A lacks a definition of the “*”; in the meantime, the significance marker of “*” in Figure 5E needs to be re-marked on the images.
10. This study indicated that CLP36 silencing could lead to the suppression on the proliferation and metastasis potential of lymphoma cells and the enhancement on the apoptosis, which was proven to be associated with the targeting on PI3K/AKT/CREB pathway (Line 209-212). Please explain how this conclusion can be reached. This conclusion seems a bit far-fetched.

Experimental design

no comment

Validity of the findings

no comment

·

Basic reporting

The manuscript is very well designed and written, being of notable relevance to the academic community. However, there are some improvements that need to be made in order to further enhance its quality.
For example, the reference section should be reviewed and updated. There are more recent studies than those cited in the manuscript. The reviewer suggests that Figure 6 be improved with more information about the work and transformed into a graphical abstract

Experimental design

The research question in the manuscript should be better explored in the abstract and introduction, which would bring greater clarity and visibility to the manuscript's objectives.

The conclusion, both in the manuscript and in the abstract, could be refined. Here is the reviewer’s suggestion: In summary, our study demonstrated that CLP36/PI3K/AKT/CREB axis is involved in the progression and development of lymphoma. This suggests that therapeutic strategies targeting the CLP36/PI3K/AKT/CREB axis may be effective for lymphoma intervention. These conclusions are based on cellular studies, and future in vivo animal studies will be conducted to further validate and complete our findings

Moreover, a paragraph describing the limitations of the study would be highly valuable for the manuscript's acceptability. You could use the last sentence of the conclusion and add other limitations.

Validity of the findings

The results are very well presented and conclusive. I would like to congratulate for the quality of the immunoblotting runs.

---

## Round 0.2 · accepted · Accept

Your revised manuscript has been reviewed by all three referees, who unanimously recommend acceptance of your paper. The reviewers agree that you have adequately addressed their previous concerns and improved the quality of the manuscript.

Reviewer 1 ·

Basic reporting

In this study, the authors demonstrated that CLP36 is a risk factor for the PI3K/AKT/CREB signaling pathway in the progression of lymphoma. Overexpression of CLP36 in lymphoma is associated with poor prognosis. The silencing of CLP36 inhibits the survival and proliferation of tumor cells, and affects the phosphorylation of PI3K, AKT, and CREB proteins in the PI3K/AKT/CREB signaling pathway. The experimental design of this study is reasonable, the argument is clear, and the evidence is sufficient. In my opinion, this article is of great significance for studying the molecular mechanism of A and developing potential therapeutic targets. After revision, the manuscript has been further improved.

Experimental design

no comment

Validity of the findings

no comment

Reviewer 2 ·

Basic reporting

no comment

Experimental design

no comment

Validity of the findings

no comment

Additional comments

This study investigated the role of CLP36 (PDLIM1) in lymphoma based on cell experiments. The expression of CLP36 is relatively high in lymphoma, which is associated with poor prognosis. In addition, CLP36 is highly expressed in lymphoma cells, and silencing CLP36 helps to inhibit the survival and proliferation of lymphoma cells, as well as enhance their apoptosis. In addition, CLP36 silencing inhibited the expression of cadherin 2 (CDH2) and vimentin (VIM) in lymphoma cells, but promoted the expression of Bax and Caspase 3, while reducing the phosphorylation of PI3K, AKT, and CREB. Overall, the experimental design of this study is rigorous and logically sound, which is a direction of interest to readers. The current version meets the publishing standards.

·

Basic reporting

no comment

Experimental design

no comment

Validity of the findings

no comment

Additional comments

no comment